# Recent Advances in the Decontamination and Upgrading of Waste Plastic Pyrolysis Products: An Overview

**Salma Belbessai** [1,2], **Abir Azara** [1,2] and **Nicolas Abatzoglou** [1,*]

1   Department of Chemical & Biotechnological Engineering, Université de Sherbrooke, Sherbrooke, QC J1K 2R1, Canada; salma.belbessai@usherbrooke.ca (S.B.); abir.azara@usherbrooke.ca (A.A.)
2   Laboratoire de Valorisation des Énergies Fossiles, École National Polytechnique, 10 Avenue Hassen Badi El Harrach, P.O. Box 182, Alger 16200, Algeria
*   Correspondence: nicolas.abatzoglou@usherbrooke.ca

**Abstract:** Extensive research on the production of energy and valuable materials from plastic waste using pyrolysis has been widely conducted during recent years. Succeeding in demonstrating the sustainability of this technology economically and technologically at an industrial scale is a great challenge. In most cases, crude pyrolysis products cannot be used directly for several reasons, including the presence of contaminants. This is confirmed by recent studies, using advanced characterization techniques such as two-dimensional gas chromatography. Thus, to overcome these limitations, post-treatment methods, such as dechlorination, distillation, catalytic upgrading and hydroprocessing, are required. Moreover, the integration of pyrolysis units into conventional refineries is only possible if the waste plastic is pre-treated, which involves sorting, washing and dehalogenation. The different studies examined in this review showed that the distillation of plastic pyrolysis oil allows the control of the carbon distribution of different fractions. The hydroprocessing of pyrolytic oil gives promising results in terms of reducing contaminants, such as chlorine, by one order of magnitude. Recent developments in plastic waste and pyrolysis product characterization methods are also reported in this review. The application of pyrolysis for energy generation or added-value material production determines the economic sustainability of the process.

**Keywords:** plastic pyrolysis; contamination; pre-treatment; products upgrading; pyrolysis applications

## 1. Introduction

Plastics have become an important part of modern life as they are ubiquitous. The demand for plastics is increasing worldwide; 367 million tons of plastics were produced in 2020 [1]. Plastic waste management is now a major concern in many countries. As an example, Canada uses 4.6 million metric tons of plastic each year, only 9% of which is recycled [2]; the rest ends up in landfills. Unfortunately, 79% of used plastic worldwide goes to landfill sites or the natural environment [3] because mechanical recycling does not tolerate mixed and contaminated plastics. Furthermore, mechanical recycling delays final disposal rather than avoiding it [3]. To this day, polyethylene terephthalate (PET) is considered the only polymer for which an efficient mechanical recycling scheme is established [4]. The degradation of plastic takes many years, creating severe risks to organisms and the environment. For instance, some animals, especially sea animals, mistake plastic for food and die from entanglement [5]. Moreover, plastic exposed to heat can decompose to greenhouse gases [6]. Plastic that ends up in oceans and rivers decomposes, releasing toxic chemical compounds that can be transferred to the human body via contaminated seafood [7].

During the COVID-19 pandemic, single-use plastic-product consumption and release have increased remarkably. For instance, personal protection equipment, such as face shields, isolation gowns, hair and shoe nets and safety glasses, comprise 72% polypropylene (PP) [8]. In addition, studies revealed that if 1% of used masks are disposed of

improperly, 40 tons of masks per month are dispersed into the environment [9]. Medical devices are composed of 10% high-temperature plastics, 20% engineered plastics and 70% commodity plastics (polyethylene [PE], polystyrene [PS], polyvinylchloride [PVC] and PP) [9]. Therefore, hospital waste management is a growing concern. Still, plastic waste comes from different sources, such as municipal solid waste (MSW), hospital waste (HW), automobile shredder residue (ASR) and waste electrical and electronic equipment (WEEE), as shown in Figure 1. PE, PP, PVC, PS and PET are the main plastics present in MSW; more plastics with further additives such as acrylonitrile butadiene styrene (ABS) and poly(methyl methacrylate) (PMMA) are present in WEEE and ASR, respectively.

Thermal destruction is one possible solution that has gained attention in recent years. Because of their hydrocarbon-based nature, waste plastics can be transformed into valuable products such as fuels. Specifically, pyrolysis is a thermochemical treatment that is considered a promising alternative for the energetic and material valorization of plastic waste. Pyrolysis converts mixed and contaminated plastics into gas, liquid and solid streams using heat in the absence of oxygen. Since plastic wastes are derived from fossil-fuel sources, their composition is similar to petrochemical fuel, and they have high calorific value; thus, they can be considered a valuable source of energy [10]. A techno-economic study by Al-salam et al. [11] revealed that the calorific value of waste is amongst the most sensitive parameters that affect the economic performance of thermal treatment processes. In addition, the pyrolysis process is a flexible technology because the operating conditions can be optimized to maximize the production of the targeted stream. Compared to gasification, pyrolysis systems produce significantly more olefin products [12]. These unsaturated hydrocarbons can be re-polymerized to produce new recycled plastics, thus closing the loop of a true cyclic economy.

Nevertheless, the production of chemical materials and fuels from heterogeneous mixtures of waste is a significant challenge. The process should consider the composition of the feedstock, which is variable. For example, the heterogeneity of the feedstock was the main reason for the failure of the Rwe-ConTherm plant (Hamm) [13]. The process was affected by corrosion, resulting in the collapse of the chimney in 2009. In addition, the pyrolysis of waste plastic produces a large spectrum of aliphatic and aromatic hydrocarbons with different molecular weights, which in most cases, can not be used without further post-treatment. Moreover, plastic waste contains various contaminants, such as halogens, metals and additives, which can be present in pyrolysis products. These contaminants are responsible for many problems, including corrosion, catalyst poisoning and clogging [14]. These challenges are why the production of high-value products through plastic waste pyrolysis has not taken off industrially.

Consequently, many studies [14–16] emphasized the need to upgrade pyrolysis products to meet the specification standards of current precursors (e.g., steam cracking) or final products (e.g., fuels). Pre-treatment, such as dehalogenation, is also required to produce streams with low contaminant concentrations. Upgrades include the cracking of long-chain molecules, reforming, separation operations and decontamination.

Several reviews have examined a variety of aspects of the pyrolysis of waste plastics. Sharuddin et al. [17] reported the influence of process parameters and different plastics on oil production and quality. They also reviewed the physical and chemical characteristics of pyrolytic oils. Chen et al. [18] reported on technologies for MSW pyrolysis focusing on reactors, products and measures to mitigate the environmental impact. In another study, Miandad et al. [19] reviewed catalytic pyrolysis in terms of pyrolytic oil composition as well as the influence of operating parameters such as temperature and retention time on the pyrolysis process. Kinetic studies, along with the techno-economic evaluation, were discussed by Kunwar et al. [20]. These studies summarized the state-of-the-art approaches to waste plastic pyrolysis, especially in terms of pyrolytic oil production. However, the information on contamination and the need to upgrade pyrolysis products is rarely discussed in the open literature. Another relevant, recent review by Kusenber et al. [14] described the contaminant composition of post-consumer plastic waste pyrolysis oil and its implication

for steam cracking. The study found that the contaminant level exceeded established limits by one or more orders of magnitude, concluding that intermediate upgrading steps are necessary to convert waste plastic into valuable chemicals through pyrolysis.

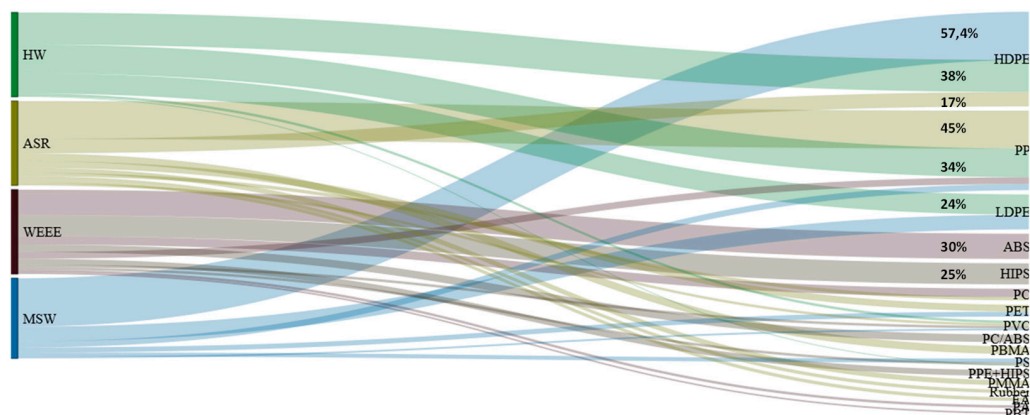

**Figure 1.** Sankey chart depicting the polymeric composition of different waste streams. Data were gathered from [21–24], and this distribution varies from country to country. HIPS: High impact polystyrene, PC: Polycarbonate, PBMA: Poly butyl methacrylate, EA: Ethyl acrylate, PA: Polyamide, PBT: Butylene terephthalate.

This work reviews the solutions proposed in the literature to upgrade plastic waste pyrolysis products. To the best of our knowledge, the information on these pre- and post-treatment methods has not been discussed thoroughly in previous reviews. Moreover, we discuss the effects of catalysts on pyrolysis products and contaminants. Finally, this paper shows the value of the final application of the gaseous, liquid and solid products. The intended uses of the products are key factors in determining whether the process is economically sustainable in a commercial application.

## 2. Pyrolysis Products and Contaminants: The Need for Pre- and Post-Treatment

### 2.1. Thermal Pyrolysis

Thermal pyrolysis is a thermochemical treatment that can be conducted at a wide range of temperatures (350–900 °C). Unlike combustion, which requires excess oxidizing agents or gasification, which occurs under stoichiometric conditions, pyrolysis is conducted in the absence of oxygen. Macromolecular polymers are decomposed into smaller molecules, forming various hydrocarbons, and oxygenates from plastics that contain oxygen (i.e., PET). Pyrolysis can be carried out at different temperature levels: low (<400 °C), medium (400–600 °C) and high (>600 °C). The temperature level and residence time define the type of pyrolysis and the desired products, as illustrated in Table 1.

**Table 1.** Pyrolysis processes based on operating conditions and targeted products adapted from [8,25].

| Process | Heating Rate | Residence Time | Temperature (°C) | Major Products |
|---|---|---|---|---|
| Slow carbonization | Very low | days | 450–600 | Charcoal |
| Slow pyrolysis | <5 °C/s | 10–60 min | 450–600 | Char, oil |
| Fast pyrolysis | 10–200 °C/s | 0.5–5 s | 550–650 | Oil |
| Flash pyrolysis | 1000 °C/s | <1 s | 450–900 | Oil, gas |

Table 2 shows the main results of experimental investigations on the pyrolysis products and their dependence on the plastic waste composition. The products were characterized using advanced analytical techniques, such as two-dimensional gas chromatography [15,26]. The product compositions vary significantly depending on the plastic waste material and type of pyrolysis (e.g., fast or slow). In slow pyrolysis, polyolefins produce large amounts

of paraffins and olefins, whereas high concentrations of aromatics and gases are generated during fast pyrolysis.

**Table 2.** Effect of plastic waste composition on the oil quality and the contaminants present in the oil.

| Feedstock | Reactor and Operating Conditions | Product Distribution (wt%) | PIONA (wt%) | Main Contaminants |
|---|---|---|---|---|
| Post-consumer plastic waste (~88% PP, ~12% PE) [15] | Continuous stirred tank reactor (CSTR) 450 °C Atmospheric pressure Feeding rate: 1 kg/h | Liquid (wax): 87 Gas: 9 Solid: 3 | Liquid oil n-Parraffins:3.1 α-Olefins: 6 Diolefins: 19.5 Aromatics: 1 Isoparaffins: 4.7 Iso-olefins: 62.7 Naphthenes: 3 Gas: NR | Fe: 21 ppmw Na: 114 ppmw Pb: 6 ppmw Si: 43 ppmw Cl: 137 ppmw |
| Post-consumer plastic waste (~46% PP, ~53% PE and ~1% others) [15] | Continuous stirred tank reactor (CSTR) 450 °C Atmospheric pressure Feeding rate: 1 kg/h | Liquid (wax): 89 Gas: 7 Solid: 3 | Liquid oil n-Parraffins: 14 α-Olefins: 12.9 Diolefins: 7.6 Aromatics: 13.6 Isoparaffins: 5.8 Iso-olefins: 39 Naphthenes: 7.1 Gas: NR | Ca: 17 ppmw Na: 82 ppmw Pb: 5 ppmw Si: 28 ppmw Cl: 474 ppmw |
| Post-consumer plastic waste (~1% PP, ~97% PE and ~2% others) [15] | Continuous stirred tank reactor (CSTR) 450 °C Atmospheric pressure Feeding rate: 1 kg/h | Liquid (wax): 85 Gas: 10 Solid: 5 | Liquid oil n-Parraffins: 34.4 α-Olefins: 25.5 Diolefins: 4.3 Aromatics: 3.9 Isoparaffins: 6.5 Iso-olefins: 13.8 Naphthenes: 11.6 Gas: NR | Fe: 3 ppmw Na: 82 ppmw Pb: 4 ppmw Si: 47 ppmw O: 2100 ppmw Cl: 143 ppmw |
| Plastic solid waste (PE, PP, PS and PA) and traces of food residuals [26] | Fast pyrolysis, 430 °C, Atmospheric pressure, Vapour residence time: 1 s. | NR | Liquid oil n-Parraffins: 5 α-Olefins: 12.3 Isoparaffins: 8.2 Aromatics: 67.1 Gas n-Parraffins: 22.3 Isoparaffins: 27.6 Naphthenes: 21.0 Aromatics: 27.1 | Nitrogen-containing compounds: 6.4 wt% Sulfur-containing compounds 0.6 wt% Oxygen-containing compounds 2.5 wt% |

NR: not reported. PIONA: paraffinic, iso-paraffinic, olefinic, naphthenic and aromatic content.

The type of plastic waste is critical when specific products are targeted. As an example, PS decomposes at low temperatures with a high yield in oil that is rich in styrene [27]. However, PE decomposes to wax at low temperatures and gas and oil at high temperatures [17]. Compared to PE, PP produces more of its monomer. As shown in Table 2, a feedstock rich in PP produces high amounts of iso-olefins and diolefins, whereas a feedstock rich in PE produces liquids containing high concentrations of linear paraffins and olefins. This is explained by the tertiary carbon present in PP, which makes the C–C bond less stable and easy to degrade. PP and PE pyrolysis follow the random scission decomposition mechanism. Therefore, a wide range of molecules, following a Gaussian distribution [10], is produced. The fragmentation of the polymeric chains produces free radicals, which can react in different ways. The most likely reaction is β-scission, producing a new free radical and an unsaturated end. The radical can also capture a hydrogen atom, creating

another radical and a saturated end, known as "intermolecular hydrogen transfer". A similar transfer can occur in the same radical, referred to as "intramolecular hydrogen transfer". This mechanism ends when two radicals meet [28]. This reaction mechanism explains the high concentration of paraffins and olefins in PE and PP pyrolysis oils. Some polymers, such as PMMA, decompose into their monomers as the polymeric chain undergoes β-scission to produce methyl methacrylate [29]; the decomposition process is known as unzipping. PS decomposes according to two mechanisms: random scission and unzipping. The most common product is styrene, accompanied by small amounts of its dimer and trimer [28]. The third decomposition mechanism is lateral-group scission, as in the case of PVC. The degradation starts with the removal of HCl from the main polymer chain. The resulting unsaturated chain turns into aromatics, such as benzene, toluene and naphthalene [30]. This divergence in the pyrolysis products requires further upgrading to make them suitable for chemical processing. For instance, when the feedstock is rich in PP, the oil produced is rich in olefins and diolefins. As it is, this oil cannot be valorized as fuel before hydrotreatment. However, if these unsaturated compounds can be removed, they become a suitable feedstock for the petrochemical industry. Table 3 shows the difference in hydrocarbon composition between plastic pyrolysis oil (PPO), vacuum gas oil (VGO), light cycle oil (LCO) and steam cracker feedstock (presented here by naphtha fraction). PPO is very rich in olefins (almost 60 wt%), which explains why it cannot be used directly as fuel or as steam cracker feedstock for monomer recovery.

**Table 3.** Composition and contaminants present in pyrolytic oil, VGO, LCO and stream cracker feedstock.

| Elements | Plastic Pyrolysis Oil (PPO) [15] | Vacuum Gas Oil (VGO) [31] | Light Cycle Oil (LCO) [32] | Steam Cracker Feedstock |
|---|---|---|---|---|
| Hydrocarbons (wt%) | | | | |
| Paraffins | 19.8 | 8.49 | 22.3 | 41.7 [1] |
| Olefins | 59.5 | - | - | - |
| Naphthenes | 7.1 | 29.16 | 15.9 | 46.2 [1] |
| Aromatics | 13.6 | 62.34 | 61.8 | 12.1 [1] |
| Contaminants (wt%) | | | | |
| S | 0.0046 | 1.17 | 0.1771 | 0.5 [2] |
| N | 0.1143 | 0.23 | 0.1375 | Light feedstock: 0.01 [2], heavy feedstock: 0.2 |
| O | <0.1 | NR | NR | 0.1 [2] |
| Other contaminants (ppm) | | | | |
| Cl | 474 | NR | NR | 3 [2] |
| Si | 28 | NR | NR | 1 [2] |
| Na | 82 | NR | NR | 0.125 [2] |

[1] Values of naphtha composition from [33]. [2] Values from [14]. NR: not reported.

Plastic wastes from different sources contain various hazardous substances that end up in the pyrolysis products. The presence of volatile chlorine and sulfur in the feedstock leads to the formation of HCl and $H_2S$ in the gaseous stream and even the liquid products [18]. Kusenberg et al. [15] reported that pyrolysis processes yield a significant reduction in the heteroatom and metal concentration in the resulting liquid phase. The majority of the heteroatoms are found in the gaseous phase, while metals are concentrated in solid carbonaceous products. Table 4 illustrates the elemental composition of solid waste (PP ~46% PP, ~53% PE and ~1% others) and its pyrolytic oil. This indicates that most of these contaminants remain in the solid residue. However, the remaining fraction of contaminants in PPO is still problematic. A recent study by Kusenberg et al. [34] confirmed the necessity of decontaminating pyrolysis products prior to steam cracking. These researchers studied the steam cracking of PPO blended with fossil naphtha. They compared the obtained yields with those of pure naphtha steam cracking. Steam cracking of PPO/naphtha yielded ~23% of ethylene at 820 °C and ~28% at 850 °C, exceeding pure naphtha's yields at both conditions (~22 and ~27%, respectively). Nevertheless, high coke formation and heat

exchanger fouling was observed with PPO/naphtha blend. This was attributed to the presence of heteroatoms and metal contaminants in plastic waste.

**Table 4.** Elemental composition of solid waste and pyrolysis oil [15].

| Element | N | S | O | Cl | Al | Ca | Cu | Fe | K | Mg | Mn | Na | Si | Zn |
|---------|-----|------|------|------|-------|--------|------|-----|-------|-------|-----|-------|------|------|
| Units | | (wt%) | | | | | | | (ppmw) | | | | | |
| Solid waste | 0.2 | <0.1 | 0.3 | 3600 | 387.5 | 1599.7 | 22.3 | 120 | 158.1 | 139.0 | 0.3 | 254.5 | 80.8 | 33.3 |
| Pyrolysis oil | 0.1143 | 0.0046 | <0.1 | 474 | 273.5 | 16.6 | 2.1 | - | 36.2 | 54.1 | 0.2 | 82.1 | 27.6 | 4.6 |

According to Table 3, sulfur and nitrogen are not problematic for PPO. Nevertheless, amounts of oxygen, chlorine, iron, sodium and silicon in the pyrolysis oil exceed the threshold values for industrial steam crackers [15]. These contaminants come from different sources: residual paper, biomass and additives. Additionally, Table 2 shows that the concentrations of these contaminants vary with the type of feedstock. More iron and sodium (21 and 114 ppmw, respectively) are found in a PPO of rich PP feedstock, while the highest concentration of oxygen (2100 ppmw) is in a feedstock rich in PE, which indicates PET contamination. This shows that part of these contaminants comes from the polymeric matrix and the contamination from products contained in plastic packaging (e.g., soap or food). Therefore, the current sorting and washing steps do not remove such elements completely. Toraman et al. [26] reported that the oxygenated compounds are in the form of (ketones, phenols, aldehydes and esters, while nitrogen comes in various forms, such as nitriles, pyridines, quinolines, indole and caprolactam, and sulfated compounds are in forms of thiols/sulfides, thiophenes/disulfides, benzothiophenes and dibenzothiophenes.

The origin of these contaminants (O, Cl, Fe, Na, Si) and the problems they cause are discussed in detail in a recently published review [14]. They are known to cause issues such as corrosion, clogging and downstream catalytic poisoning [14]. If the pyrolysis oils were used as fuels, these elements might trigger undesirable reactions and cause gum formation [32].

*2.2. Catalytic Pyrolysis: The Effect of Catalyst on Pyrolysis Products and Contaminats*

Catalytic pyrolysis has been tested at different scales with various types of plastic streams. The use of a catalyst in pyrolysis decreases the activation energy of the process, thus accelerating the reaction rate. This saves energy as the operating temperature is reduced. In catalytic pyrolysis, the C–C bonds of the polymers are broken on Brønsted acidic sites of the catalyst. Moreover, the catalyst offers better selectivity toward specific products and improves their quality [35]. Catalysts can be in contact with the plastic (in situ catalytic pyrolysis) or in a two-step process (thermal pyrolysis followed by catalytic cracking), also referred to as in-line pyrolysis or ex situ catalytic pyrolysis [36,37]. This last configuration is more advantageous as the temperature of pyrolysis and catalytic upgrading can be controlled independently [38]. In addition, the catalyst is more efficient, and its deactivation is delayed [39] as the poisoning of acid sites by the inorganic contaminants and asphaltenes/heavy waxes is reduced. Most inorganic contaminants contained in the plastic waste are expected to stay in the char inside the pyrolysis reactor, which can be removed occasionally [40]. Both homogeneous (i.e., one liquid phase) and heterogeneous (i.e., solid phase) catalysts have been used in plastic pyrolysis. The most well-known homogeneous catalysts are Lewis acids, such as $AlCl_3$ [41]. However, heterogeneous catalysts are the most commonly used for plastic pyrolysis because the catalyst can be separated from the products and recovered. The most common heterogeneous catalysts are classified as nanocrystalline zeolites; conventional solid acids, such as zeolites; fluid catalytic cracking (FCC) catalysts silica-alumina; mesostructured catalysts, such as MCM-41; and metal supported on basic oxides [17,20,42].

Zeolites are crystalline aluminosilicates, consisting of a sequence of $SiO_4$ and $AlO_4$ units; the ratio $SiO_2/Al_2O_3$ determines the type of zeolite and its reactivity [43]. Zeolites

have been widely studied in the catalytic pyrolysis of waste polymers as one of the most effective solid catalysts for the cracking of plastic waste [44]. Generally, the use of zeolites leads to an increased yield of volatiles [45]. Both the $SiO_2$–$Al_2O_3$ ratio and the pore size of zeolites have significant influences on pyrolysis products and catalyst deactivation. Elordi et al. [46] reported that HZSM-5, having the smallest pores, was more selective to $C_2$–$C_4$ olefins with a yield of 60 wt% (29% propene, 21% butenes and 10% ethane), compared to HY-zeolite and Hβ-zeolite. Coke deposition on HZSM-5 was less than that on the other zeolite catalysts because the growth of coke precursors in zeolites with larger pores gave rise to polyaromatic structures that remain inside the pores, owing to hindered counter-diffusion. Meanwhile, HZSM-5 micropores caused a steric hindrance that limited bimolecular hydrogen transfer. Consequently, HZSM-5 deactivation was not significant when compared to deactivations by Hβ-zeolite and HY-zeolite. This phenomenon was confirmed by similar studies [47,48]. Moreover, Miskolczi et al. [49] indicated that HZSM-5 has the highest activity in double-bond isomerization in municipal plastic waste (MPW) oil, as listed in Table 5. In their study, the HZSM-5 catalyst increased the concentration of internally positioned double bonds from 17.7% to 66.9%. In addition, HZSM-5 showed excellent efficiency in oil deoxygenation and aromatic hydrocarbon formation. Oxygenated products are undesirable in PPO. They increase the oil viscosity and decrease its heating value and stability while rendering the PPO corrosive [50]. When blending PP with PC, the concentration of oxygenates in the presence of HZSM-5 was reduced from 72.3% to 2.9% [51]. By the effects of both Brønsted and Lewis acid sites, the alkenes and alkanes produced from PP trigger aromatization reactions (cyclization, Diels–Alder, dehydrogenation and hydrogen transfer reaction) [52]. These reactions provide hydrogen radicals, which are contacted with oxygenates (phenols, ethers and furans) from PC. In addition, HZSM-5 promotes the direct hydrodeoxygenation of adsorbed phenols by dehydration [51,53]. In the presence of enough light hydrocarbons, HZSM-5 can also promote Diels–Alder reactions of benzofurans into aromatic hydrocarbons [54].

**Table 5.** Effect of catalyst on pyrolysis products and contaminants. Concentrations are as specified in the respective reference.

| Feedstock | Catalyst/Sorbent | Reactor and Operating Conditions | PIONA | Undesired Elements/Compounds in PPO (ppm) | Relevant Remarks |
|---|---|---|---|---|---|
| Municipal plastic waste (MPW) Miskolczi et al. [50,55] | No catalyst | Batch reactor, 500 °C, ratio of catalyst to MPW: 1/10 | 20% paraffins 23% olefins | S: 51 Cl: 618 Ca: 297 Zn: 124 Br: 253 Sb: 105 | Presence of 926 ppm of Cl and 520 ppm of Br in the gas |
| | Y-zeolite | | 11.5% paraffins 18% olefins | S: 34 Cl: 457 Ca: 282 Zn: 146 Br: 194 Sb: 99 | Presence of 1355 ppm of Cl and 594 ppm of Br in the gas |
| | β-zeolite | | 4.5% paraffins 9.8% olefins | S: 37 Cl: 399 Ca: 273 Zn: 128 Br: 201 Sb: 114 | Presence of 1291 ppm of Cl and 601 ppm of Br in the gas |
| | FCC | | NR | S: 44 Cl: 422 Ca: 291 Zn: 117 Br: 205 Sb: 128 | Presence of 1166 ppm of Cl and 552 ppm of Br in the gas |

**Table 5.** *Cont.*

| Feedstock | Catalyst/Sorbent | Reactor and Operating Conditions | PIONA | Undesired Elements/Compounds in PPO (ppm) | Relevant Remarks |
|---|---|---|---|---|---|
| | $MoO_3$ | | 22.2% paraffins 25% olefins | S: 42 Cl: 451 Ca: 299 Zn: 140 Br: 185 Sb: 91 | Presence of 1352 ppm of Cl and 596 ppm of Br in the gas |
| | Ni-Mo-catalyst | | 15% paraffins 26.8% olefins | S: 39 Cl: 416 Ca: 281 Zn: 129 Br: 219 Sb: 113 | Presence of 1403 ppm of Cl and 591 ppm of Br in the gas |
| | HZSM-5 | | 18.5% paraffins 23.7% olefins | S: 42 Cl: 487 Ca: 304 Zn: 132 Br: 266 Sb: 104 | Presence of 1210 ppm of Cl and 555 ppm of Br in the gas |
| | $Al(OH)_3$ | | 10% paraffins 15% olefins | S: 29 Cl: 372 Ca: 295 Zn: 127 Br: 201 Sb: 97 | Presence of 594 ppm of Cl and 407 ppm of Br in the gas |
| PP/PE/PS/PVC/ABS-Br (3/3/2/1/1) Brebu et al. [56] | No catalyst | Single-step fixed-bed reactor, 450 °C | High amounts of aromatics More than 50% of PPO is benzene derivatives (n-C8 n-C10) | Cl: 4972 Br: 1924 N: 1214 | Bromine compounds: bromomethane, bromobutane, bromophenol and dibromophenol |
| | α-FeOOH | | | Cl: 3370 Br: 170 N: 840 | More effective in Br removal |
| | Fe-C | | | Cl: 1014 Br: 170 N: 981 | Faster degradation and highest amount of oil (67 wt%) |
| | Ca-C | | | Cl: 113 Br: 418 N: 1370 | More effective in Cl removal |
| | $CaCO_3$ | | | Cl: 355 Br: 1161 N: 1078 | More effective in Cl removal |
| MPW Lopez-Urionabarrenechea et al. [57] | No catalyst | Semi-batch reactor, 440 °C | NR | Cl in liquid: 0.2% Cl in gas: 5.3% Cl in solid: <0.1% | |
| | ZSM-5 | Conventional catalytic pyrolysis Semi-batch reactor, 440 °C | 95.1% aromatics 2.8% olefins | Cl in liquid: 1.2% Cl in gas: 1% Cl in solid: 0.4% | 81.5% of C5–C9 compounds |
| | | Stepwise pyrolysis, 300 °C for 60 min then 440 °C | 80.6% aromatic 4.8% olefins | Cl in liquid: 0.3% Cl in gas: 3% Cl in solid: 0.4% | 74.4% of C5-C9 compounds More >C13 compounds Loss of catalyst activity |
| | | Non-catalytic dechlorination + catalytic pyrolysis | 94.2% aromatics 3.3% olefins | Cl in liquid: 0.3% Cl in gas: 2.2% Cl in solid: 0.4% | 82.0% of C5–C9 compounds |
| PC/PP (1/3) Sun et al. [51] | HZSM-5 | Two-staged tubular furnace, 500 °C | 95.8% aromatics 4.2% oxygenates | Phenols: 4.2% Furans: 0% Ethers: 0% | The aromatics yield reached 98.1% at 700 °C The presence of PP improved the deoxygenation effect of oxygenate compounds |

**Table 5.** *Cont.*

| Feedstock | Catalyst/Sorbent | Reactor and Operating Conditions | PIONA | Undesired Elements/Compounds in PPO (ppm) | Relevant Remarks |
|---|---|---|---|---|---|
| MPW Miskolczi and Ates [45] | No catalyst | Stirred batch reactor, 500 °C | 32.8% paraffins 49.5% olefins 9.7% aromatics 4.0% naphthenes 4.0% oxygenates | Cl: 1285 Br: 1533 P: 498 S: 71 Sb: 189 | Oil density (at 20 °C): 0.848 g/cm$^3$ Oil viscosity at 40 °C: 133 mPa·s |
| | β-zeolite | | 31.8% paraffins 47.3% olefins 3.5% aromatics 4.3% naphthenes 3.1% oxygenates | Cl: 1273 Br: 1563 P: 574 S: 51 Sb: 179 | Oil density (at 20 °C): 0.814 g/cm$^3$ Oil viscosity at 40 °C: 113 mPa·s High efficiency in increasing volatile yields |
| | y-zeolite | | 32.0% paraffins 49.1% olefins 3.1% aromatics 4.5% naphthenes 3.0% oxygenates | Cl: 1322 Br: 1407 P: 663 S: 57 Sb: 173 | Oil density, g/cm$^3$ (at 20 °C): 0.822 Oil viscosity at 40 °C, mPas: 119 |
| | Ni-Mo-catalysts | | 31.4% paraffins 49.0% olefins 2.8% aromatics 5.8% naphthenes 2.8% oxygenates | Cl: 1135 Br: 1522 P: 582 S: 65 Sb: 164 | Oil density (at 20 °C): 0.828 g/cm$^3$ Oil viscosity at 40 °C: 126 mPa·s Increases H2 production |
| MPW + heavy oil (1/3) Miskolczi and Ates [45] | No catalyst | Stirred batch reactor, 500 °C | 34.9% paraffins 52.1% olefins 9.1% aromatics 1.8% naphthenes 2.1% oxygenates | Cl: 173 Br: 264 P: 115 S: 16 Sb: 47 | Oil density (at 20 °C): 0.832 g/cm$^3$ Oil viscosity at 40 °C: 216 mPa·s |
| | β-zeolite | | 27.2% paraffins 47.4% olefins 9.8% aromatics 4.1% naphthenes 2.0% oxygenates | Cl: 210 Br: 385 P: 117 S: 14 Sb: 43 | Oil density, g/cm$^3$ (at 20 °C): 0.782 Oil viscosity at 40 °C: 168 mPa·s |
| | y-zeolite | | 30.1% paraffins 46.7% olefins 11.5% aromatics 2.7% naphthenes 2.3% oxygenates | Cl: 214 Br: 326 P: 94 S: 15 Sb: 37 | Oil density (at 20 °C): 0.787 g/cm$^3$ Oil viscosity at 40 °C: 181 mPa·s |
| | Ni-Mo-catalysts | | 34.6% paraffins 46.2% olefins 9.0% aromatics 4.0 naphthenes 2.4% oxygenates | Cl: 195 Br: 279 P: 102 S: 9 Sb: 51 | Oil density (at 20 °C): 0.792 g/cm$^3$ Oil viscosity at 40 °C, mPas: 202 |

Regarding the contaminants, Table 5 shows that most of the elements (S, Cl, Ca, Zn, Br and Sb) were found in the PPO when the gas phase had only S, Cl and Br contaminants, which was caused by the dehalogenation reactions and the formation of HCl and HBr [58]. This shift in halogens was intensified during catalytic pyrolysis; catalysts decreased the chlorine and bromine content in PPO and increased their respective amounts in the gas phase. The catalytic pyrolysis also reduced the concentration of other contaminants (Ca, Zn, Sb) compared to thermal pyrolysis, although no significant difference was observed among the different catalysts in terms of decontamination efficiency. Owing to its alkalinity, Al(OH)$_3$ was the most efficient in removing acidic contaminants in PPO. Table 5 also shows that the Ca–C composite was more effective in chlorine removal with a 97% reduction. Nonetheless, these results are from different studies with different reactors and operating conditions.

Lopez-Urionabarrenechea's study [57], described in Table 5, recommended the following configuration when the feedstock contains PVC: a low-temperature dechlorination step complemented with alkaline additives to capture HCl, followed by a catalytic step at higher temperatures to avoid the loss of catalyst activity during the dechlorination step. In

another study [45], iron-based catalysts (α-FeOOH and Fe-C) tested on pyrolysis of a mix of plastic containing ABS-Br were found to be effective in removing more than 90 wt% of bromine from PPO. Nonetheless, these catalysts have small effects on the removal of organic nitrogen (20–30 wt%). Table 5 also shows that the co-pyrolysis of MPW and heavy oil with a mass ratio of 1/3 could significantly decrease the concentration of contaminants in the resulting PPO by one order of magnitude. These results show the potential of reducing contamination by diluting plastic waste or its derived PPO in petroleum-based feedstock.

Fluid catalytic cracking (FCC) catalysts are composed mostly of Y-zeolite crystals, activated alumina and kaolinite [59]. Their complex compositions make them suitable for a variety of cracking reactions. FCC catalysts are mainly used in the petroleum industry to upgrade the heavy fraction of crude oil into light fractions, such as gasoline. The FCC catalyst that is used in plastic pyrolysis is often a spent catalyst, commonly referred to as an equilibrated FCC catalyst [42]. Fortunately, this catalyst has no cost, and it is a waste material from the petroleum industry. Studies show that an FCC catalyst still has cracking ability despite the contamination from previous usage [60,61]. Due to the reduced acidity, this catalyst generates a much lower coke yield compared to a fresh FCC catalyst. The acidity is lowered because of the poisoning of active sites by metal contaminants [40]. Consequently, this catalyst is not effective in contaminant removal, as shown in Table 5.

The effect of catalytic pyrolysis on the product decomposition and distribution has been extensively studied with different configurations [62,63], reactors [38,40], catalysts [55,56] and operating conditions [64]. Nevertheless, the catalyst effect on contaminants has only been investigated minimally, to the best of our knowledge. Therefore, more investigations are required to study the behaviour of catalysts in real-world plastic waste. The effect of different contaminants on the catalyst performance and the contribution of catalysts in the decontamination process are also areas to explore.

### 2.3. Advances in Characterization of Waste Plastic and Pyrolysis Products

Plastic waste stream consists of a mixture of different polymers containing several sources of contaminates such as paper, food residue and metals. Characterizing waste stream is critical for waste management. Contaminates identification is important for identifying the recycling route for waste plastic. There are several techniques to characterize the plastic waste, including differential scanning calorimetry (DSC), Fourier transform infrared (FTIR) spectroscopy and inductively coupled plasma optical emission spectroscopy (ICP-OES) or mass spectroscopy (ICP-MS).

ICP-OES and ICP-MS are used to determine metal concentrations in the polymeric waste. This method detects ultra-trace (ppb) of metal concentration [65]. Roosen et al. [66] performed the ICP-OES analysis of different plastic packaging waste. They found that the highest concentrations in Fe (270 ppm), Zn (45.6 ppm) and Mg (186 ppm) are attributed to PP and PS packaging trays. For the determination of halogens and sulfur concentrations, combustion ion chromatography (CIC) is used [67,68]. In this method, the sample is firstly pyrolyzed in an oxidizing atmosphere; the resulting vapours are absorbed by an adequate absorbent and then introduced to the IC system for separation and quantification. This method is advantageous because it contains an automated sample preparation for both solids and liquids. The C, H, N, S and O composition of waste plastic is usually detected using an elemental analyzer [15]. Thermogravimetric analysis (TGA) is also used to investigate the thermal behaviour of the plastic waste [68]. Nowadays, the use of coupling techniques such as TG-MS, TG-FTIR [69,70] and TG-FTIR-MS [71] to analyze the degradation of waste plastic through characterization of the resulting products, is getting more popular. In this context, some researchers used Pyro-GC (Pyrolyzer-gas chromatography) to investigate the fast pyrolysis of plastic waste by analyzing the quality of the products [72]. Plastic waste is also analyzed by FTIR to identify functional groups, organic, polymeric and inorganic materials [73].

One of the most used techniques for pyrolysis products analysis is gas chromatography coupled with different detectors such as flame ionization detector (FID) for quantifying

hydrocarbons; electron capture detection (ECD) for halogenated hydrocarbons; thermal conductivity detector (TCD) for $CO_2$, CO, $H_2$, $O_2$, Ar, $N_2$ analysis; MS for functional group, aromatics and double bond analysis.

Recently, a remarkable progress is achieved in identifying pyrolysis oil components due to a powerful technique, which is two-dimensional gas chromatography (GC × GC). This method provides more detailed information on the composition, compared to one-dimensional GC. The GC × GC uses two columns of different stationary phases, connected by a modulation tool. There are two types of modulation: thermal and flux modulator. The effluent passes through both columns, which creates tow retention times for each component. One detector is enough for the analysis, though several detectors can be used to take profit from their advantages [74], as it is illustrated in Table 6. The two columns are of different polarities, when the first one is nonpolar and the second is polar the arrangement is called normal phase (NP), when it is the inverse it is called reverse phase (RP).

**Table 6.** GC × GC coupled with different detectors, for the characterization of PPO.

| 2D-GC Technique | Column Arrangement | Columns Used | Molecules Detected | Reference |
|---|---|---|---|---|
| GC × GC-FID | NP | Two plot columns: PTMSP poly-(1-trimethylsily-1-propyne) GASPRO silica | Saturated and unsaturated hydrocarbons from C3–C8 | [75] |
| GC × GC-FID | NP | RTX-1 PONA (Dimethyl polysiloxane) BPX-50 (50% phenyl polysilphenylene-siloxane) | Diolefins, iso-olefins, mononaphthenes, n-paraffins, iso-paraffins and monoaromatics from diesel fraction | [26,34,76,77] |
| | RP | Stabilwax (polyethylene glycol) Rxi-5 ms (5% diphenyl 95% dimethyl polysiloxane) | Diolefins, iso-olefins, mononaphthenes, n-paraffins, iso-paraffins and monoaromatics from diesel fraction | [77] |
| GC × GC-NCD (nitrogen chemiluminescence detector) | NP | RTX-1 PONA (Dimethyl polysiloxane) BPX-50 (50% phenyl polysilphenylene-siloxane) | Nitrogen compounds | [15,26] |
| GC × GC-SCD (sulfur chemiluminescence detector) | NP | RTX-1 PONA (Dimethyl polysiloxane) BPX-50 (50% phenyl polysilphenylene-siloxane) | Sulfur compounds | [15,26] |
| GC × GC-ToF-MS (time of flight MS) | RP | RTX-1 PONA (Dimethyl polysiloxane) BPX-50 (50% phenyl polysilphenylene-siloxane) | Oxygenated compounds | [26] |

## 3. Waste Plastic Pre-Treatment

### 3.1. Plastic Separation

Plastics such as PVC and PET, which are present in MPW, produce dangerous substances during pyrolysis. HCl and chlorinated hydrocarbons, such as chloroform ($CHCl_3$) and dichloromethane ($CH_2Cl_2$), are formed from PVC [78]. These organic and inorganic chlorides can corrode the pyrolysis equipment, contaminate other products and cause air pollution without appropriate gas-emission control modules [14,78]. Moreover, if these chlorinated hydrocarbons are oxidized (burned), more harmful products, such as dioxins and furans, can result [79]. PET thermal decomposition leads to the formation of carbonic acids, such as benzoic and terephthalic acids, which are problematic to the pyrolysis facil-

ity, causing corrosion and clogging in the piping [80] (terephthalic acid is solid at room temperature). In addition, the pyrolysis of PET is less interesting than its mechanical or chemical recycling (e.g., hydrolysis, methanolysis, glycolysis, ammonolysis and aminolysis). Chemical recycling leads to the complete depolymerization of PET [81]. Consequently, the separation of mixed plastics is required before pyrolysis.

The various methods of plastic separation include manual separation, gravity separation by flotation [79,82], centrifugal separation [83], triboelectrostatic separation [84] and selective dissolution [85]. Manual separation is inefficient and labour-intensive; gravity separation is limited by the similar specific gravities of plastics, such as PVC (1.3–1.4) and PET (1.38–1.41) [84]. In triboelectrostatic separation, the tribo-charger imparts the charge on the plastic particles, for example, in a fluidized bed. The particles are charged negatively or positively depending on their work function (i.e., their relative affinity for electrons). Then, the particles are separated in an electrostatic separator where they can be deflected towards the appropriate counter electrode [84]. This method is more effective when the difference in the work functions of plastics is significantly high. The selective dissolution method consists of dissolving plastics in a solvent that targets only one polymer. The dissolved plastic is recovered by the rapid evaporation of the solvent [86] or by the addition of an appropriate "anti-solvent" to make the polymer precipitate [85]. As the solvents are toxic and expensive, this method is rather inconvenient.

For MPW, the pyrolysis is preceded by drying to reduce the moisture content before entering the reactor. The heat required by the dryer mainly comes from the combustion of part of the pyrolysis products [87]. Pre-treatment also includes size reduction by crushing and sieving the plastic, especially when working with fluidized bed reactors (FBRs).

### 3.2. Dehalogenation

The chlorine content in pyrolysis products is related to the presence of PVC, while bromine comes mainly from brominated flame retardants in ABS and HIPS. WEEE plastics are principally composed of HIPS, ABS, PVC and PC, as illustrated in Figure 1, which makes them rich in halogens. As an example, the pyrolysis of ABS releases different brominated products such as HBr, $CH_3Br$, $C_2H_5Br$, $C_3H_5Br$, $C_3H_7Br$ and $C_3H_5BrO$ [88]. Some researchers also reported the presence of bromophenol and dibromophenol during the pyrolysis of WEEE [89]. In order to obtain fuels or chemical products from WEEE, dehalogenation treatment is required prior, during or after pyrolysis [23]. In the literature, dehalogenation is focused on plastics rich in halogens, such as PVC and flame retardant plastics [90]. However, some researchers [14,15] concluded that the dehalogenation treatment of plastic waste, even polyolefin waste, is necessary for the PPO to meet current specifications set for steam cracker feedstock. A high level of chlorine in PPO can come from the PVC contamination of the plastic waste or from the adsorption of salt that was in the packaged product [91]. Thus, we illustrate some examples of debromination and dechlorination in the following paragraphs.

Cagnetta et al. [92] investigated the dehalogenation of PP containing the flame retardant decabromodiphenyl ethane (DecaBDE) by means of mechanochemical pre-treatment. The debromination of PP was carried out using Fe-SiO$_2$ or CaO-SiO$_2$ in a planetary ball mill at room temperature. After eight hours of dry milling, 90% of the bromide was recovered when using Fe-SiO$_2$, and 80% was the recovery of bromine in the case of CaO-SiO$_2$. Organic bromine contained in PP was mineralized into soluble inorganic bromide. With the high-energy milling and presence of SiO$_2$, iron particles become smaller and activated. These fine iron particles with high surface energy become electron donors [23]. The transfer of electrons to the flame retardant occurs according to the following equation [92]:

$$F + C_{12}Br_{10}O \rightarrow C_{12}Br_9O^{\cdot} + Br^- + Fe^+ \tag{1}$$

This reaction continues until the debromination and carbonization of DecaBDE [93]. The polymeric chain also captures the electrons from iron particles, which causes cleavages. This facilitates the next processing step of pyrolysis [92,93].

Grause et al. [94] studied the removal of the same flame retardant from HIPS using NaOH/ethylene glycol solution (NaOH(EG)) in both a stirred flask and a ball mill reactor between 150 and 190 °C. The debromination reached 42% in the stirred flask at 190 °C and about 98% in the ball mill reactor after 24 h. Therefore, ball milling had a more positive effect on the debromination process. Analytical methods showed that debromination was achieved by the substitution of bromine by hydroxyl groups (from NaOH) or hydrogen (from DecaBDE) [95,96]. The reaction is controlled by diffusion in both the stirred flask and ball mill reactor with an activation energy of 205 kJ mol$^{-1}$. The polymer matrix did not change; it was just cross-linked through the DecaBDE backbone.

As mentioned previously, when the feedstock contains PVC, a dechlorination process is needed to reduce the chlorine content in the pyrolysis products. There are several methods of dechlorination, such as stepwise pyrolysis (i.e., two-step pyrolysis), catalytic pyrolysis or the addition of adsorbents in the feedstock. In stepwise pyrolysis, the plastic is heated at a low temperature to decompose PVC and capture HCl; this step is called dehydrochlorination. In the second stage, the remainder of the plastic is heated to a high temperature. López et al. [97] performed the stepwise pyrolysis of a mixture of plastics containing PVC at different temperature and time conditions, the addition of CaCO$_3$ and the combination of both methods. They reported that 300 °C and 60 min were the optimum conditions in the dechlorination step to reduce the liquid chlorine content by 50 wt%. However, the authors noticed that stepwise pyrolysis led to the formation of heavy hydrocarbons and fewer aromatics. The addition of CaCO$_3$ was efficient in capturing HCl and reducing chlorine content in the gases significantly (to 0.9 wt%). Nevertheless, the concentration of chlorine in the liquid (0.6 wt%) was higher compared to the result of the stepwise pyrolysis (0.2 wt%). The combination of both methods led to lower HCl generation, but the liquid chlorine content was the same as in stepwise pyrolysis. The efficiency of stepwise pyrolysis for dechlorination was also reported by another study where 90% of chlorine was recovered as HCl in the dechlorination step at 350 °C for 60 min [98].

Recent studies investigated the efficiency of hydrothermal treatment for the chlorine removal of waste feedstock. The main advantage of this technology compared to other dechlorination methods is the enhancement of heat and mass transfer due to the homogeneous reaction. The supercritical or subcritical water present in the system works simultaneously as a solvent and a catalyst for acid-catalyzed reactions [99]. Li et al. [100] carried out the hydrothermal treatment of pure PVC in a batch reactor. The highest dechlorination efficiency of 94.3 wt% was obtained at 240 °C with 1% NaOH. Wang et al. [101] studied the effect of hydrothermal dechlorination pre-treatment on oil production through the fast pyrolysis of mixed plastics. Results showed that the dechlorination efficiency reached 99.9 wt%, and the total yield of oil and wax increased by 7.06 wt% after pyrolysis. Furthermore, methane selectivity increased by 17.81%, owing to the possible weakening of the C–C bond energy of the β-position during the hydrothermal pretreatment.

Nishibata et al. [102] investigated the effect of superheated steam with catalysts and adsorbents on the simultaneous dechlorination and degradation of PVC. They have found that the CaO caused more dechlorination and degradation than other metal oxides, including Fe$_3$O$_4$, SiO$_2$, Al$_2$O, Ca(OH)$_2$ and MgO, in the presence of superheated steam. The temperature is increased by the exothermic reaction of CaO with steam, which promotes PVC degradation. The newly formed HCl reacts with CaO and Ca(OH)$_2$ to form calcium chlorides such as CaCl$_2$ and CaClOH. After degradation in the presence of CaO and steam, 91 wt% of chlorine present in the sample was found in the inorganic phase.

Most industrial applications use inexpensive alkaline additives, such as calcium oxide and sodium carbonate, in the plastic feedstock to remove HCl [103]. They also employ an alkaline solution to wash the gas in a scrubber to remove all acids from the stream [104]. Agilyx [103], which uses stepwise pyrolysis, is the only current technology capable of handling plastic waste containing up to 70% PVC. In the first step, the plastic is heated under a vacuum inside a batch reactor, during which the moisture and HCl are

separated from the feedstock [105]. Additionally, BASF in Germany pursues a two-step technology [42]. The dehydrochlorination step is carried out at 250–380 °C. The system handles feedstock with PVC content lower than 5%. The HCl produced is recycled for PVC production.

## 4. Products Upgrading

### 4.1. Dittillation of Pyrolytic Oil

The oil obtained from the fast pyrolysis of waste plastic is usually a dark liquid composed of various hydrocarbon compounds from $C_5$ to $C_{30}$. Although fractional distillation is frequently used in the petroleum industry, information on pyrolytic oil distillation is scarce. Some researchers used distillation to split pyrolytic oil into gasoline, diesel and heavy oil fractions for fuel recovery. Others performed distillation to recover monomers and close the loop toward new virgin plastic. For instance, Baena-González et al. [106] carried out the distillation of PPO at atmospheric pressure up to 240 °C. This operation led to a bitumen at the bottom of the column and a distilled fraction. The resulting bitumen contained aromatics (55.05 wt%) and saturates (33.41 wt%). The detailed composition of the bitumen indicated its potential to be added in asphalt or bituminous mixtures. The distilled fraction was also rich in aromatics (54.72 wt%), with styrene as the principal compound, followed by ethylbenzene and toluene. These results indicate that the feedstock contained high amounts of PS and that the pyrolysis conditions favoured aromatization reactions. The distilled fraction was subjected to a liquid–liquid extraction with sulfolane to separate aromatic compounds from other components. Another fractional distillation was carried out to separate the different aromatic compounds and recover styrene (73.26 wt%). This study demonstrated the technical feasibility of producing different materials including bitumen, olefins, toluene and styrene from the fractionalization of PPO.

Thahir et al. [107] studied the pyrolysis of waste PP in a pyrolysis reactor integrated with a distillation bubble cap plate column (Figure 2) to optimize liquid products. Experiments were conducted using 500 g of plastic waste. Vapours produced from pyrolysis of waste plastic flow through the column. Ash residue and wax stays in the reactor, whereas non-condensed vapour flows through the riser to reach the cap and eventually, forms liquid bubble (mixture of vapour and condensate). The pyrolysis temperature affected the liquid fuel characterization yielded on each tray of the column, as described in Table 7. The total liquid oil yield at 500–560 °C reached 88 wt% with the highest yield of gasoline (67 wt%). However, at 650 °C, the diesel yield reached 83 wt%. This study shows the possibility of tuning the pyrolysis temperature to optimize the desired fuel. The physicochemical characteristics of these fuels, such as density, viscosity, octane-cetane number, ash content and calorific value, are similar to those of conventional fossil fuels. However, the chemical composition was not reported.

**Table 7.** Distribution of fuel products along the column [107].

| Temperature (°C) | Plate I | Plate II | Plate III | Plate IV |
|---|---|---|---|---|
| 500–560 | Gasoline | Gasoline | - | - |
| 580–600 | Kerosene | Gasoline | Gasoline | - |
| 620–650 | Diesel-wax | Kerosene | Gasoline | Gasoline |

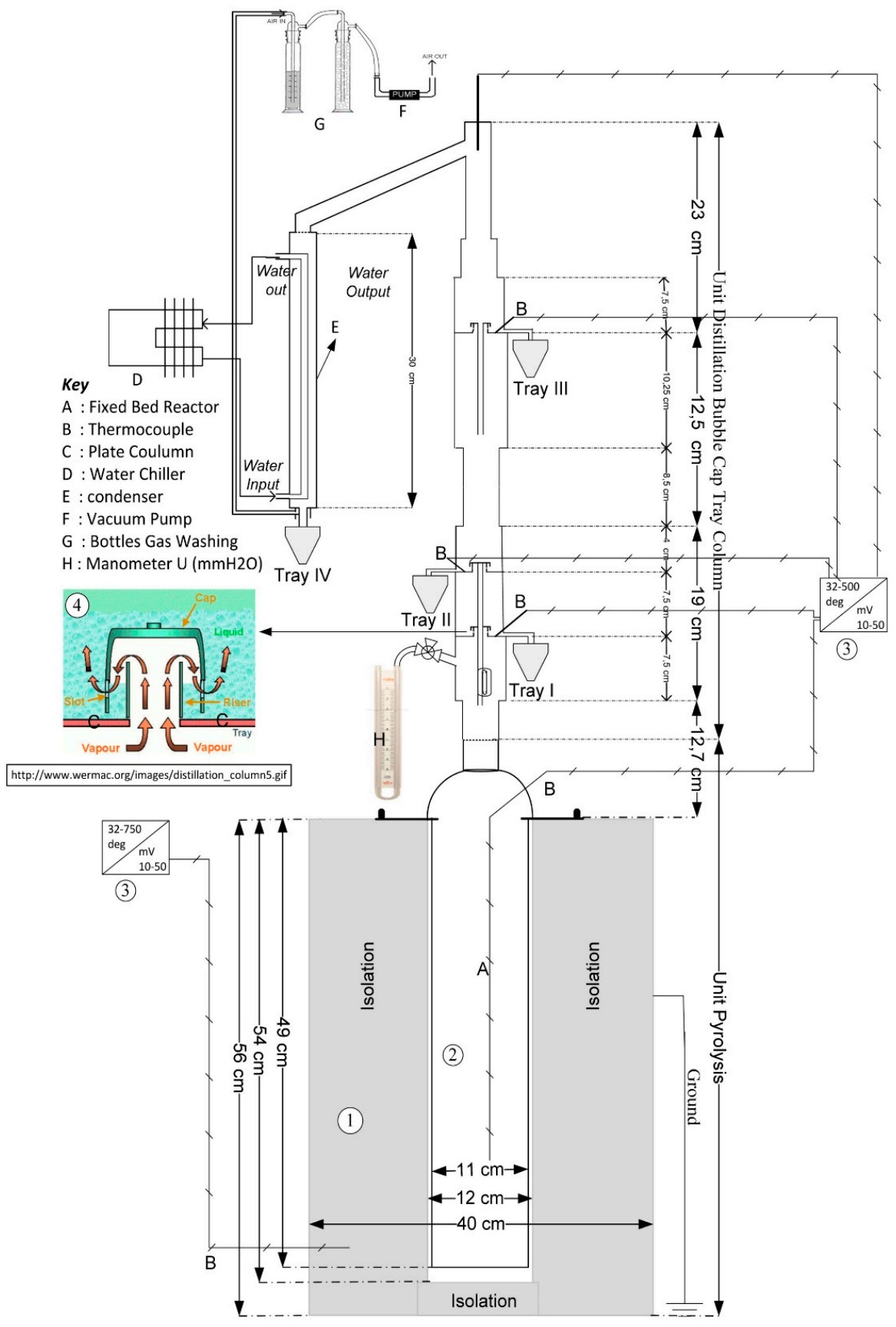

**Figure 2.** Schematic diagram of thermal pyrolysis integrated with a distillation bubble cap column, reproduced with permission [107].

Another study that investigated the distillation of PPO was carried out by Lee et al. [108]. The PPO came from a Korean pyrolysis kiln facility that treats ten tons/day of mixed plastic waste at approximately 450 °C. The objective was to collect pyrolysis oil fractions similar to petroleum diesel based on carbon number. First, atmospheric distillation was performed to recover the specific fractions following the boiling points of different petroleum fuels (~169 °C for gasoline, 138–278 °C for kerosene and 138–399 °C for diesel). Then, vacuum distillation was conducted to reduce the heat duty. At 100 °C lower than that for atmospheric distillation, similar carbon fractions were obtained in distillation at vacuum conditions. The main yields of the different fractions are gathered in Figure 3.

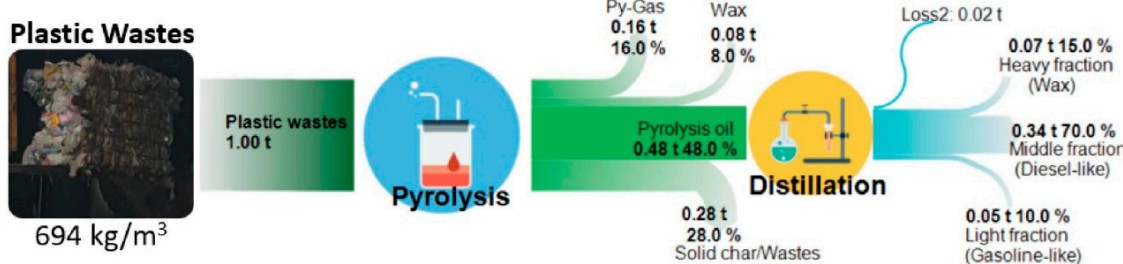

**Figure 3.** Plastic waste pyrolysis process mass balance from Lee et al., reproduced with permission [108].

Most commercial pyrolysis plants continuously fractionate the liquid product to control the carbon distribution of the different fractions [42].

Dao Thi et al. [77] performed a detailed group-type characterization of both naphtha ($C_5$–$C_{11}$) and diesel fractions ($C_7$–$C_{23}$) originating from the distillation of PPO by means of two-dimensional gas chromatography. Table 8 shows that both fractions were rich in olefins and diolefins, which indicated that further processing would be required, such as hydroprocessing, because, as mentioned before, high contents of unsaturated compounds negatively affect the quality of the fuels, owing to the gum formation through secondary reactions [109]. The original PPO contained high amounts of aromatics (67.1 wt%), while the naphtha and diesel fractions had an aromatic content of 9 and 2 wt%, respectively. Therefore, the aromatics present in PPO had a high carbon number, and they remained at the bottom of the distillation column. The presence of heteroatoms (S, N, O) in both fractions was reduced compared to PPO. The fractionalization led to low concentrations of heteroatom-containing compounds in both light and heavy fractions.

**Table 8.** PIONA and elemental composition of naphtha, diesel and PPO determined by comprehensive two-dimensional gas chromatography analysis [77].

| Elements | Naphtha | Diesel | PPO [a] |
|---|---|---|---|
| | **PIONA (wt%)** | | |
| Paraffins | 15 | 28 | 5 |
| Isoparaffins | 2 | 4 | 8.2 |
| α-olefins | 35 | 36 | 12.3 |
| Iso-olefins | 9 | 9 | - |
| Diolefins | 4 | 4 | - |
| Naphthenes | 26 | 17 | - |
| Aromatics | 9 | 2 | 67.1 |
| | Elemental composition (wt%) | | |
| C | 85.93 | 85.51 | 88 |
| H | 13.93 | 14.49 | 10.9 |
| S | 0.021 | 0.001 | 0.17 |
| N | 0.003 | ND | 1.06 |
| O | 0.14 | 0.01 | 0.35 |

ND: not detected. [a] the composition of PPO is taken from a previous study of the same research group [26].

*4.2. Pyrolysis Wax Treatment in FCC Units*

The cracking of polyolefinic pyrolysis waxes in an FCC unit has been studied extensively [110]. This cracking is adopted for recovering raw materials and obtaining fuels. Studies have shown that the cracking of waxes leads to higher yields of gasoline compared to the cracking of VGO [111]. Rodríguez et al. [31] investigated the FCC of HDPE pyrolysis waxes in a riser simulator reactor under industrial conditions in order to produce fuels from waxes coming from a pyrolysis plant. The waxes were obtained during a fast pyrolysis of HDPE at 500 °C in a conical spouted bed reactor. The reaction of FCC was carried out at temperatures from 500–560 °C, catalyst/oil mass ratios of C/O = 3–7 and a residence time of 6 s, which are typical values used in the industry. Conversion values of HDPE waxes varied from 36.7–5.1 wt%, increasing when the temperature and catalyst–oil ratio increased. The yield, which grouped following the distribution used in refineries, at 530 °C and C/O = 5 was the following: dry gas (4 wt%), liquefied petroleum gas (LPG) (14 wt%), naphtha (28 wt%), light cracked products (LCO) (43 wt%), heavier cracked products (HCO) (7 wt%), coke (4 wt%). Olefins were the most abundant hydrocarbons in the naphtha fraction, followed by aromatics, isoparaffins, n-paraffins and naphthenes. The temperature and C/O had substantial effects on the product distribution. High cracking temperatures increased the paraffinic fraction and reduced the aromatics.

Some authors discussed the possibility of integrating pyrolysis plants with refineries [42,112]. Liquid wax derived from the pyrolysis of waste plastic can be fed, along with oil products, into steam reforming, hydroprocessing, FCC and coking processes for fuel production, as illustrated in Figure 4. Monomers and light hydrocarbons can be directed to petrochemical plants for the production of new polymer resins [113]. This recycling configuration allows the valorization of all kinds of plastic waste and their pyrolysis products, while minimizing the landfilled fraction.

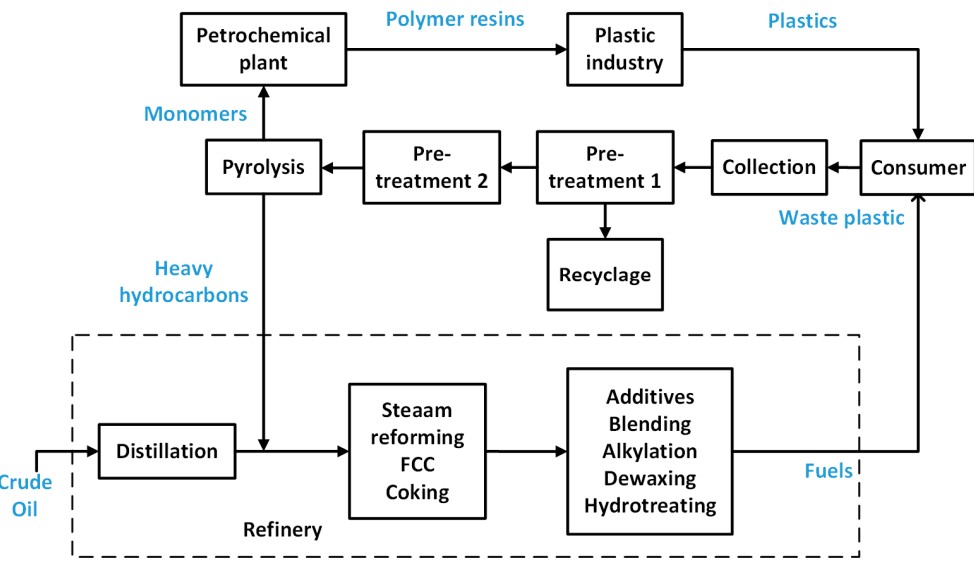

**Figure 4.** Integration of a plastic waste pyrolysis plant with an oil refinery.

In this context, Rodríguez et al. [114] complemented their study of HDPE pyrolysis waxes in an FCC unit, but this time they used a blend of HDPE waxes and VGO (1:4 mass ratio). The reaction was carried out in a laboratory-scale reactor mimicking the behaviour of an industrial FCC reactor. The results showed that the conversion values of the blend at 500 and 530 °C (40.6–47.6 and 49.3–55.5 wt%, respectively) were slightly lower than those of pure VGO (41.4–47.3 and 51.1–55.5 wt%, respectively). Nevertheless, at 560 °C, the blend showed a higher conversion (63.1–66.3 wt%) compared to the VGO (61.1–62.7 wt%) because the cracking of the waxes was promoted at high temperatures. The yields of naphtha and LPG increased with the blending, whereas that of dry gas decreased.

### 4.3. Catalytic Upgrading of Pyrolysis Liquids

The use of a catalyst can improve the pyrolysis liquids by breaking the long hydrocarbon chains and increasing the selectivity of the desired products. A catalyst can be used in the pyrolysis process, as explained in Section 2.2, or as a post-treatment for upgrading the liquid phase. Lee et al. [115] studied the effects of zeolites on catalytic upgrading of pyrolysis wax oil. This oil was obtained from the pyrolysis of MPW in a commercial rotary kiln reactor. The catalytic experiments of wax upgrading were conducted in a continuous fixed-bed reactor at 450 °C using three commercial zeolites: HZSM-5, HY-zeolite and modernite (HM). The HZSM-5 zeolite gave the highest gas yield (51.04 wt%) compared to the other zeolites, with a selectivity toward aromatic and cyclic components. HY showed medium catalytic activity with high paraffinic content, and the carbon number of these was between 5 and 6. The HM catalyst, having a one-dimensional pore structure, showed the lowest catalytic activity.

Furthermore, Wang et al. [116] designed a practical laboratory pyrolysis oil catalytic separator, which is a combination of distillation and catalytic cracking (Figure 5). The oil was from an MPW pyrolysis company, and the catalysts used were zeolite 4A and Cu-(MDC-7) and Ni-based catalysts. The temperature was kept between 320 and 380 °C, and the products were separated into three categories: F1 (gasoline-like fraction), F2 (diesel-like fraction) and F3 (wax). The results showed that the presence of catalysts decreased the mass yield of F2, owing to the loss of some gases (e.g., CO, $CO_2$, $CH_4$) through decarboxylation, decarbonylation and dehydration reactions. Compared to other catalysts, MDC-7 generated the highest mass yield of F1 (15.8 wt%), whereas the highest yield of F2 (66.3 wt%) was produced with Ni-based catalyst. The use of catalysts reduced the heavy carbon range (>$C_{23}$) from 22.1 wt% in the original oil to 0.1–1.6 wt% in F1 and 7.3–8.4 wt% in F2. Moreover, F1 and F2 fractions from catalytic separation had lower total acid number (TAN) values compared to those of thermal separation, suggesting that more deoxygenation reactions took place in the presence of catalysts. The order of deoxygenation capacity was Ni-based catalyst followed by MDC-7 and zeolite 4A. In terms of composition, MDC-7 exhibited high aromatic and naphthenic contents, while the Ni-based catalyst showed the highest content of olefins in F1 (54.49%) and F2 (36.16%). The authors suggested that the catalytic reaction mechanisms of both catalysts were as follows:

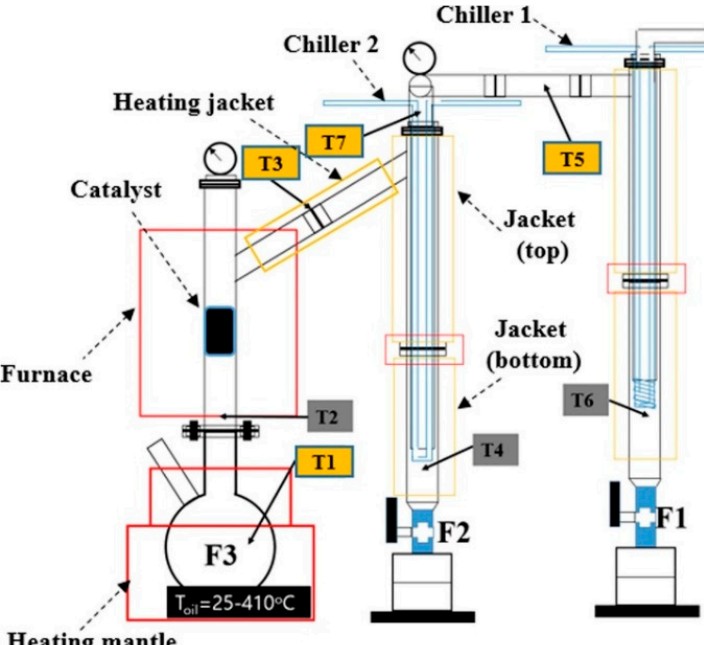

**Figure 5.** Schematic diagram of a laboratory pyrolysis oil separation system, reproduced with permission from [116].

For the MDC-7 catalyst, long-chain paraffin, olefins, alcohols and ester were converted into short-chain olefins and paraffins through catalytic cracking, decarboxylation, decarbonylation and dehydration. Then, aromatics and naphthenes were formed via aromatization and cyclization, respectively.

- With the Ni-based catalyst, long-chain alcohols were transferred into olefins via dehydration and catalytic cracking, while benzoic acid and phenols were transferred into aromatics through deoxygenation reactions. Some long-chain paraffins were cracked into short-chain paraffins.

### 4.4. Pyrolysis Oil Hydroprocessing

As previously mentioned, the composition of the pyrolytic liquid can vary depending on the feedstock and may contain undesirable compounds. Hydrotreating the liquids can help reduce the olefins and aromatics and remove heteroatoms (N, S, Cl and O). Hydrodenitrogenation occurs according to the following reaction [117]:

$$C_5H_5N + 5H_2 \rightarrow C_5H_{12} + NH_3 \tag{2}$$

The hydrodechlorination of chlorobenzene as a prevalent chlorinated hydrocarbon is shown in the reaction below [14]:

$$C_6H_5Cl + H_2 \leftrightarrow C_6H_6 + HCl \tag{3}$$

Hydrodeoxygenation of an oxygenated compound is schematically presented below [117]:

$$R - OH + H_2 \rightarrow R - H + H_2O \tag{4}$$

This operation is conducted in the presence of a hydrotreating catalyst at temperatures ranging from 190–340 °C and pressures of 20–204 atm [104]. These conditions help achieve the removal of heteroatoms while the cracking is minimized. Ding et al. [118] used bifunctional (acidic/metallic functions) catalysts (NiO/HBeta, NiO/HSAPO-11 and NiO/HMCM-41) for the hydrocracking of waxes obtained from the pyrolysis of polyolefins at 300 °C in a stirred autoclave reactor, under 20 atm of hydrogen. With a mixture of Ni/H-Beta and ZSM-5, the hydrocracking led to higher fractions of gases (30.2 wt%) and diesel (23.5 wt%). A comparison of the catalysts showed that the ratio of acid-to-metal function sites affects the mechanism of hydrocracking and hydroisomerization of waxes. When the acid strength was high and the hydrogenation power of the catalysts was weak, more isoparaffins and lighter hydrocarbons were produced, which decreased the oil pour point. On the contrary, when the catalyst had a higher metal function, the hydrogenation of olefins was the predominant reaction, which lowered the production of isoparaffins. Therefore, the authors suggested a two-stage reactor system with the use of both catalysts to have a medium composition.

Moreover, hydrocracking allows us to tailor the selectivity toward the desired fuel by adjusting the temperature, as shown in Table 9. Higher temperatures favour the end-chain cracking; thus, more light hydrocarbons are produced. The PPO nature is also an important factor; for example, when PP pyrolysis oil, which is rich in olefins, goes through complete hydrogenation, large amounts of saturated hydrocarbons are produced [119]. A comparison of the physiochemical properties of this oil, the hydrogenated oil and diesel is outlined in Table 10. Hydrogenation enhanced the density, viscosity, cetane index, flash point, fire point and pour point. The properties of the hydrogenated oil matched the EN590 standards. This hydrogenated PP oil was blended with diesel, and promising results were obtained during engine performance trials. This application will be presented in more detail in Section 5.1.

**Table 9.** Main results from hydrotreatment of different pyrolysis oil.

| Feedstock | Reactor Type | Pressure (atm) | Catalyst | Temperature (°C) | Main Results |
|---|---|---|---|---|---|
| LDPE pyrolysis oil (~47.7 wt% gasoline (C5–C12), ~36.2 wt% light diesel (C13–C18), ~16.1 wt% heavy diesel (C$_{19}$–C$_{40}$) [120] | Stirred autoclave reactor | 20 | Pd/h-ZSM-5 | 250 | Reduction of gasoline fraction through oligomerization and increasing the share of light and heavy diesel up to 41.8 wt% and 20.3 wt%, respectively |
|  |  |  |  | 310 | Light diesel decreases to 26.2 wt% High production of isoparaffins (34.5 wt%) through hydroisomerization |
|  |  |  |  | 350 | Cracking is dominant, producing 11.5 wt% of gases and 56.6 wt% of gasoline High production of aromatics (24.2 wt%) |
| HDPE pyrolysis oil (26.5 wt% naphtha, 33.1 wt% LCO, 40.4 wt% HCO) [121] | Stirred tank reactor (STR) in semi-batch regime | 80 | NiW/HY | 400 | Product distribution: LCO (~28 wt%), naphtha (~29 wt%) and gas (~10.4 wt%) |
|  |  |  |  | 420 | Product distribution: LCO (~23.3 wt%), naphtha (~35 wt%) and gas (~30.3 wt%) |
|  |  |  |  | 440 | Product distribution: LCO (~14.3 wt%), naphtha (~30.8 wt%) and gas (45.9 wt%) Naphta rich in isoparaffins and one-ring aromatics |
| PP pyrolysis oil (67 wt% alkanes, 20 wt% alkenes and traces of aromatics) [119] | Autoclave reactor | 70 | Ni/ZSM-5 | 350 | Complete conversion of alkenes to alkanes, hydrogenated PP oil contained 97% alkanes. Alkanes distribution: 8.3 wt% (C1–C10), 63 wt% (C10–C20), 25 wt% (C20–C30) |
| LDPE pyrolysis oil (48 wt% gasoline, 35 wt% diesel and 15 wt% heavy diesel) [122] | Stirred autoclave reactor | 20 | Ni/h-ZSM-5, Ni/h-Beta, Ni/Al-MCM-41, Ni/Al-SBA-15 | 310 | Complete hydrogenation of alkenes for all catalysts except Ni/h-ZSM-5, due to its high cracking activity. |
| LDPE pyrolysis products [123] | Stirred autoclave reactor | 5–40 | Ni/h-β | 250–350 | Higher temperatures promote aromatization reactions Higher pressures promote hydrogenation of olefins and Saturation of more than 80% of olefins |
| Polyolefins pyrolysis oil [124] | Stirred autoclave reactor | 20 | Ni/h-β | 310 | Saturation of more than 90% of olefins Amount of gasoline + light diesel was within 80–85% |

Regarding heteroatom removal, Miller et al. [125] reported that with 1 wt% HZSM-5, the hydroprocessing reduced chlorine content from 50–70 ppm to 2–8 ppm. Similarly, Lingaiah et al. [126] studied the dehydrochlorination of MPW-derived oil using different catalysts: iron oxide, iron oxide-carbon composite, ZnO, MgO and red mud. The original oil contained almost 600 ppm of chlorine. After hydrotreatment, the concentration of chlorine was reduced to 32–140 ppm, with the iron oxide catalysts being the most effective and stable. However, a study on catalytic poisoning in the presence of different halogenic and metallic

contaminants is required. Metal removal techniques such as membrane filtration [127] may be needed prior to hydroprocessing.

**Table 10.** Physicochemical properties of PPO, hydrogenated PPO and diesel [119].

| Parameter | PPO | Hydrogenated PPO | Diesel |
|---|---|---|---|
| Density (kg/m$^3$) | 771.4 | 851.5 | 837.5 |
| Pour point (°C) | −30 | −20 | −15 |
| Flash point (°C) | 20 | 65 | 72 |
| Fire point (°C) | 30 | 72 | 82 |
| Calculated cetane index (N/A) | 60 | 62 | 52 |
| Kinematic viscosity at 40 °C (mm$^2$/s) | 1.78 | 3.5 | 2.31 |
| Gross calorific value (KJ/kg) | 44,957 | 44,915 | 45,593 |
| Ash content (%) | 0.01 | 0.01 | 0.01 |
| Conradson carbon residue (%) | 0.10 | 0.10 | 0.18 |

### 4.5. Fuel Properties Enhancement

If the pyrolysis liquids are saturated with paraffinic compounds, a dewaxing step is required. The presence of long-chain hydrocarbons in the fuel leads to high cloud and pour points. Acidic catalysts such as zeolites have been used for catalytic dewaxing. These catalysts have big pores, which can selectively isolate the long straight n-paraffins and crack them [104]. Dewaxing reactions are usually performed in a semi-batch system at around 450 °C and 4 atm [128].

Pyrolytic liquids are thermodynamically unstable and tend to go through polymerization and oxidation, which is mainly caused by the presence of unsaturated components. This process can lead to the formation of sediments, gums, dark colours and asphaltene agglomeration, affecting the combustion performance of the fuel. Several additives can be added to the fuel derived from waste plastic to overcome this problem and meet the required standards. Amine-based antioxidants are commonly used to prevent diesel oxidation and radical polymerization reactions [42]. The chemical compound 4-tert-butylcatechol is also used as a polymerization inhibitor in pyrolytic oil [106]. Detergents and dispersants, such as alkylphenols, are other additives that can keep oil-insoluble fractions suspended and prevent agglomeration [42].

### 4.6. Char Upgrading

Char is a by-product of the plastic pyrolysis process. It is a porous carbon material composed mainly of volatile matter and fixed carbon, but it can also contain mineral matter initially present in the feedstock [17]. High temperatures promote the formation of char [129]. During the pyrolysis of contaminated plastics, most of the contaminants stay in the char, as discussed in Section 2.1. Consequently, char cannot be used as raw material, and an upgrading process is necessary. This step can improve the process efficiency and sustainability and avoid the addition of char to landfill waste.

Bernardo et al. [130] tested the effect of the dichloromethane (DCM) extraction of char residue produced during the co-pyrolysis of a waste mixture composed of plastics (i.e., PE and PS), pine biomass and used tires. The analysis showed that DCM extraction removed organic contaminants of high to medium volatility, such as benzene, toluene, ethyl benzene, xylene (BTEX) and alkyl phenol compounds, effectively. In contrast, non-volatile organics and heavy metals (Cd, Pb, Zn, Cu, Hg and As) remained in the char and were not extracted. Subsequently, the treated char was classified as a hazardous and ecotoxic material. In a following study, Bernardo et al. [131] carried out a treatment to upgrade crude chars produced from the co-pyrolysis of different mixtures of plastics, biomass and tire waste. The chars were firstly treated by sequential organic solvent extractions with organic solvents (i.e., hexane, hexane acetone and acetone); then, they were subjected to an acidic demineralization with HCl. The results showed that the solvent extraction treatment allowed the recovery of 63–81% of the pyrolytic oil trapped in the crude

char. The demineralization procedure was efficient in the removal of 64–86% of inorganic contaminants (Al, Mg, Pb, Cr, Na, Fe, K, Mn, Mo, Ca). The resulting chars were mainly mesoporous and macroporous materials with adsorption capacities of 3.59–22.2 mg/g for methylene blue dye.

## 5. Applications

Product applications indicate whether the process is profitable on an industrial scale. One pyrolysis unit can have different applications to profit from all the products and improve the process efficiency. The unit also improves the environmental sustainability of the process by minimizing the waste that would have required disposal (char) and the use of the gases to generate energy.

### 5.1. Liquid Oil Applications

Liquid oil from pyrolysis has exhibited great potential as a new energy resource. The experimental calorific value of polyolefin-derived oil is higher than 40 MJ/kg [132], which is considered more than enough for energy utilization. The physical properties of this oil are also similar to those of commercial diesel and gasoline. However, the crude oil needs several treatments before it can be used as fuel. When liquid oil is the desired product, the optimum pyrolysis temperature ranges from 500–550 °C [133,134]. If a catalyst is used, this temperature range is lower [17]. The use of a suitable catalyst may improve the oil yield and its quality, except for PS, which yields a high liquid fraction without catalysts [135]. A recent study [106], as mentioned in Section 4.4, demonstrated that the synthesis of PS as possible using styrene recovered from PPO with a yield of 77.64% and a molecular weight of more than 53,000 g/mol. Therefore, separating PS from other plastics is recommended to recover styrene instead of extracting it from the pyrolysis oil. In this context, Zayoud et al. [76] studied the pyrolysis of used PS in a pilot-scale reactor at different pressures (0.02, 0.5 and 1.0 bara) and temperatures (450, 500, 550 and 600 °C). The objective of the study was to maximize styrene production. Authors found that 0.02 bara and 550 °C are the optimum conditions for the production of styrene with 55.9% yield. The other compounds of the liquid pyrolyzate consist of poly-aromatics that contain styrene dimers and trimers.

Some research has been conducted on the use of PPO in diesel engines. However, a comparison of the literature results is difficult, as oils derived from different plastics do not have the same composition. In most studies, blends of PPO and conventional diesel were used to avoid modifying the engine. Mangesh et al. [136] performed an experimental investigation to identify the type of plastic that gives the most suitable oil for diesel engine fuel. HDPE, LDPE, PP and PS were pyrolyzed separately, and the oil produced for each type of plastic was analyzed and compared with diesel. PP oil was selected because its physicochemical properties (e.g., density, viscosity, cetane index) most closely matched those of diesel. Engine tests were conducted on TurbochargedEicher E483 using various blends of PP pyrolysis oil (5, 10, 15%). The blends showed an ignition delay and a decrease in engine efficiency. Moreover, CO, NOx and HC emissions were significantly higher than pure diesel. A similar study was conducted by Singh et al. [137] using different ratios of non-treated mixed PPO (10, 20, 30, 40 and 50%). The results showed comparable engine efficiencies with that of diesel fuel. However, the authors also reported that the use of different blend ratios of PPO increased the exhaust emissions, owing to the presence of oxygenated compounds.

These studies reveal that the physicochemical properties of the PPO are not the only criteria for the oil to be used in the diesel engine. In the first study, PP pyrolysis oil was rich in alkenes, which increased the combustion delay and lowered the engine efficiency. The PPO used in the second study, showed better results in terms of engine efficiency, resulted from the pyrolysis of a mixture of real waste (HDPE, LDPE, PP, PS, PET and thermoset plastics). This oil was more varied in its composition in that it contained alkenes, alkanes, aromatics and 17.54% oxygenated compounds. To enhance their results, Mangesh et al. [119] performed catalytic hydrogenation on the PP oil. Details of this hydrogenation

are described in Table 8. Hydrogenated PP oil was blended with diesel in ratios of 10, 20, 30 and 40 wt%. Blends of 10 and 20 wt% showed combustion, exhaust emission and engine performance on par with pure diesel. The higher blend ratios (30 and 40 wt%) decreased the efficiency of the engine slightly and increased the CO, CO2, NOx and unburned hydrocarbon (UHC) emissions. The hydrocracking of PP pyrolysis oil yielded an oil rich in alkanes and lower in carbon number, which improved the combustion results. Nevertheless, the information regarding the economic viability of the process is lacking.

### 5.2. Solid Products

#### 5.2.1. Carbon Nanotubes

Different studies have been conducted to explore the possibilities of using pyrolysis products in different applications as materials rather than energy sources. One such application is the production of nanocarbons, such as carbon nanotubes (CNTs) and nanofilaments (CNFs), with the potential for hydrogen production [138]. These materials are higher-value products that could render the pyrolysis process more efficient and techno-economically and socio-politically sustainable. This technology mixes the appropriate catalyst with the plastic waste in one reactor, or a two-stage reactor system, where the hydrocarbons produced in the first reactor interact with the catalyst in the second reactor. Of these options, the two-step approach is recommended, which allows the regeneration of the catalyst. CNTs are produced when the gases coming from the pyrolysis of waste plastic interact with a catalyst at temperatures between 600 and 1200 °C in a chemical vapour deposition (CVD) process [139]. In this process, the carbon contained in the hydrocarbons precipitates as graphitic nanofilaments at the surface of the catalyst.

CNTs are used to reinforce polymer composites because of their mechanical and electronic properties [140]. They are valuable in many applications where electrical conductivity is critical, owing to $sp^2$ hybridization in the carbon structure [141]. To maximize the production of CNTs, the degradation of waste polymers should be promoted into light hydrocarbons and aromatics, which are efficient precursors [142]. Azara et al. [143] comprehensively described the synthesis of filamentous carbon nanomaterial via the catalytic conversion of waste plastic pyrolysis products.

Ni-based catalysts are known to have good activity for C–C and C–H cleavage, and so they are widely used for catalytic reforming to produce CNTs. Zhang et al. [138] tested the production of CNTs from waste tires using different catalysts: $Co/Al_2O_3$, $Cu/Al_2O_3$, $Fe/Al_2O_3$ and $Ni/Al_2O_3$. The results indicated that $Ni/Al_2O_3$ had the highest performance for the production of multi-walled CNTs, along with a high $H_2$ yield. Some studies suggested using a bimetallic catalyst to gain the synergic effect of the interaction between two metals. Yao et al. [144] studied the effect of a $Ni$-$Fe/Al_2O_3$ catalyst on the production of CNTs and $H_2$ from waste plastic pyrolysis. The highest $H_2$ yield of 8.47 $g_{H2}/g_{plastic}$ and the highest yield of carbon were obtained at a high loading of Fe. In contrast, at a high Ni loading, the CNTs had narrow diameters and uniform distribution.

In another study, Yao et al. [145] investigated the synthesis of multi-walled CNTs from waste plastics, using a combination of two metals, Fe and Ni, supported on four silica-alumina materials: ZSM5, MCM41, NKF5 and H-Beta. Ni-Fe/MCM41, with the largest surface area and pore size, produced the highest carbon (55.6 wt%) and $H_2$ (38.1 $mmol_{H2}/g_{plastic}$) yields. The Raman spectroscopy analysis showed that the CNTs produced from Ni-Fe/MCM41 had a more graphitic nature and fewer defects than other catalysts. Hence, the formation of Fe-Ni alloys catalyzed the growth of CNTs.

This technology has yet to be scaled up because of the challenges it faces [146]. The yield and quality of CNTs depend on several parameters, such as the type of catalyst, the reforming temperature and the shapes of the metallic particles. Moreover, some of these parameters and process variables affect the production of CNTs in an interdependent way. The heterogeneity of the feedstock and the presence of contaminants also make the formation mechanism of CNTs hard to determine. Research has demonstrated that different plastics produce different yields and qualities of CNTs [147]. Moreover, the separation of

CNTs from the catalyst must be well-defined in continuous processes. Pilot-scale systems should be developed to demonstrate the efficiency of transforming waste plastic into CNTs and hydrogen.

The production of CNTs from waste plastic is a promising way to generate high-value products, reduce their cost and promote this composite-filler technology. Furthermore, a life cycle assessment study [148] has shown that integrating CNT production with the pyrolysis process benefits the environment and decreases human toxicity and terrestrial eco-toxicity potentials.

### 5.2.2. Char

Char can potentially be used as an adsorbent for different environmental applications. Miandad et al. [149] synthesized carbon–metal double-layered oxide (C/MnCuAl-LDOs) adsorbents to study Congo red adsorption. The char used for the preparation of this adsorbent was a by-product of PS pyrolysis. The char was crushed and thermally activated in a muffle furnace at 550 °C. Then, it was chemically activated with a solution of H2SO4 and HNO3. The final adsorbent was effective for Congo red removal, with an adsorption capacity of 345.2 mg/g at pH = 4.0. Acosta et al. [150] prepared a KOH-activated carbon from tire pyrolysis char. This adsorbent eliminated bisphenol A with a capacity of 123 mg/g.

Moreover, char can be utilized for heavy metal and metalloid adsorption. Singh et al. [151] used non-modified char derived from pyrolysis of a mixture of PVC, PET and PE for arsenic adsorption. The effect of the feedstock material on char adsorption was studied. The highest-performing char for arsenic adsorption was produced from PVC and PE, which had an efficiency of 99.4%.

Furthermore, char can be used as a filler material to produce epoxy-composite materials. Sogancioglu et al. [152] studied the behaviour of char-based epoxy-composite material using PP pyrolyzed char. Chars were obtained from pyrolysis of PP at different temperatures (300 to 700 °C). With the highest aromatic content, the pyrolyzed char at 700 °C improved the hardness of the epoxy composites. Increasing the amount of char led to more epoxy-composite electrical conductivity for all chars tested. These composite materials are used in the automobile, aircraft and microelectronics industries.

Char also has the potential to be used in energy applications. Jamradloedluk and Lertsatitthanakorn [153] reported that char manufactured from HDPE has a calorific value of 4500 cal/g. To increase its surface area, the char was crushed and thermally activated at 900 °C for three hours. Then, it was extruded to produce kilogram briquettes. One briquette was able to boil water from room temperature within 13 min.

### 5.3. Gas

Gases comprise the non-condensable fraction produced from plastic waste pyrolysis. They are mainly composed of light hydrocarbons such as $H_2$, $CH_4$, $C_2H_4$, $C_2H_6$, $C_3H_8$ and $C_3H_6$ [154]. The production of gases is favoured at high temperatures and short residence times because unsaturated gases undergo secondary reactions to form aromatics [155,156]. The presence of a catalyst promotes the formation of gaseous products [19]. The gases have high calorific values between 40 and 50 MJ/kg [129]. They can be used for energy generation or in the pyrolysis system to produce energy for endothermic decomposition. Moreover, light olefins, such as ethene and propylene, are high-value monomers that can be used in the petrochemical industry after separation from other gases.

In a recent study, Eschenbacher et al. [40] tested different steam-treated industrial FCC-type catalysts and HZSM-5 additives for the in-line catalytic upgrading of pyrolysis vapours derived from PE and real (contaminated) mixed polyolefins. The purpose of the study was to maximize the production of light olefins. The severe steaming pre-treatment of the catalyst was carried out to limit the formation of coke by reducing the acidity. The steam-treated HZSM-5 additive showed the highest selectivity toward $C_2$–$C_4$ olefins, with a yield of 69 wt% (19% $C_2H_4$, 22% $C_3H_6$, 10% 1,3-$C_4H_6$ and 18% other $C_4$ olefins), obtained at high catalyst loading and temperature (700 °C). In addition, a high yield of

$C_5$–$C_{10}$ aliphatics (up to 42 wt%) was produced using the FCC catalyst. The processing of real mixed polyolefins with the HZSM-5 exhibited similar performance with even higher polypropylene production (31 wt%). The coke loads per catalyst on the steamed and unsteamed HZSM-5 were 40 and 60 $\mu g/m^2$, respectively. This study showed success with the tuning of different parameters (catalyst type, catalyst leading and temperature) to maximize the production of high value–based chemicals. Moreover, this investigation demonstrated the potential of a two-step process and a suitable catalyst to produce light monomers instead of pyrolysis oil by employing steps for upgrading (hydrotreatment and steam cracking).

To maximize the light olefin production, Santos et al. [157] designed an integrated reactor/separation system (Figure 6), where only the light hydrocarbons could leave the reactor. The pyrolysis experiments of HDPE were carried out at different temperatures (400, 450 and 500 °C) in both thermal and catalytic pyrolysis. The catalyst used was HZSM-5 with 1% (*w/w*) loading. Increasing the pyrolysis temperature led to an increase in the gas yields for both thermal and catalytic pyrolysis and the product distribution were in the range of $C_2$–$C_8$. At 500 °C with the catalyst, the gas yield reached almost 100%, with a product distribution in the range of $C_2$–$C_6$. Furthermore, the overall O/P ratio in catalytic pyrolysis was almost six times that in thermal pyrolysis. The gaseous yield could also be increased by increasing the coolant temperature at the reactor outlet, also increasing the average molecular weight of the products. This new simple design allows the conversion of HDPE into valuable short olefins that can be used in the petrochemical industry.

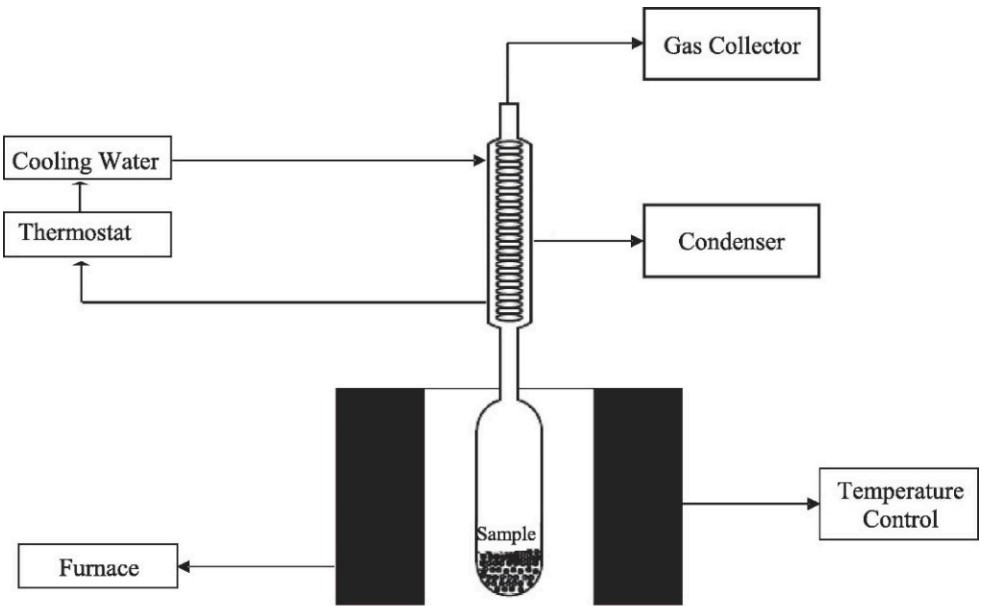

**Figure 6.** Reactor/separator set-up scheme, reproduced with permission [157].

## 6. Conclusions

Pyrolysis is a way of recovering waste plastics that cannot be mechanically recycled and will otherwise end up in the environment. The contaminants in plastic waste create challenges for the success of this technology. The use of a catalyst in pyrolysis can tailor the products for a specific application and reduce contaminants. Some alkali sorbents, such as Al(OH)$_3$, are also efficient for the removal of acidic contaminants. Pyrolysis products need further treatments either to eliminate the undesirable materials, such as HCl, or to enhance the properties of the products. Pyrolytic oil cannot be considered as a final product; therefore, the following treatments are suggested: the integration of pyrolytic plants with oil refineries, deployment of appropriate environmental safety devices and treatment of pyrolytic liquids with hydrogen-based technologies. Moreover, to achieve good-quality products, such as fuels and chemical precursors, the pretreatment of the feedstock is

necessary because satisfactory products cannot be obtained from a heterogeneous mixture of waste. Moreover, the presence of contaminants such as heteroatoms and metals lead to operational problems.

The integration of pyrolytic plants with oil refineries to process pyrolytic oil in FCC, hydrocracking and steam reforming units is necessary. This integration will lead to lower contaminant levels by dilution. Moreover, the hydroprocessing of PPO gives promising results in terms of deoxygenation and decontamination.

These additional steps increase both capital and running costs, which may lead to economic challenges. To optimize efficiency, mass and energy balances should consider all the steps involved starting from the pre-treatment of the feedstock and including all the entropic heat losses. A plausible proof of self-sustainability should also be provided to evaluate the net operational efficiency. Moreover, quality standards should be formulated to match the specifications of the current refinery feedstock.

- Pyrolysis products can be used in several applications and this targeted application determines the economic sustainability of the process:
- Upgraded pyrolytic oil can be used as fuel in diesel engines or fed to steam crackers for the production of new monomers.
- CNTs with strong mechanical and electronic properties can be produced.
- Upgraded chars can be used as adsorbents.
- Gases with high calorific values can be used for energy generation or light olefin production.

Finally, reduce and re-use strategies need to take priority with the challenges facing current recycling techniques. Governments should support pyrolysis technology to reduce waste rather than make a profit.

**Author Contributions:** Conceptualization, S.B. and N.A.; methodology, S.B.; validation, S.B., A.A. and N.A.; investigation, S.B.; resources, N.A.; data curation, S.B.; writing—original draft preparation, S.B.; writing—review and editing, N.A. and A.A.; visualization, S.B.; supervision, N.A.; project administration, N.A.; funding acquisition, N.A. All authors have read and agreed to the published version of the manuscript.

**Funding:** Funding from NSERC; PRIMA Quebec, KWI Polymers Solutions Inc. and Soleno Inc. Grant No is: RDCPJ 500331-16.

**Informed Consent Statement:** Not applicable.

**Data Availability Statement:** Not applicable.

**Conflicts of Interest:** The authors declare no conflict of interest.

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
