# Peer review of "Recent Advances in the Decontamination and Upgrading of Waste Plastic Pyrolysis Products: An Overview"

_processes, doi:10.3390/pr10040733_

Round 1

Reviewer 1 Report

The authors present an interesting review about the recent advances in decontamination and upgrading waste plastic pyrolysis products. Here I introduce some comments for the author's consideration.

  • Consider adding to the manuscript the data about the need that could be covert by producing fuels from this kind of waste.
  • Table 9 only includes a few experiments. Please, if possible, consider adding more references to this table.
  •  I miss a more detailed characterization of the obtained fuel from waste plastics.

Author Response

We thank the reviewer for the insightful comments.

See attached file.

Reviewer 2 Report

The plastic pollution and its recycling is significant. Pyrolysis is an alternative offer advantages due to its sifnificant volume reduction and energy recovery, However, how to effectively use the pyrolysis products is a chanllenge. This paper focuses on the decontanmination and upgrading of pyrolysis products. It has significant novelties seeking and can be accepted after minor revision, The comments is as follows. Regarding the dehalgenation, it is also significant for the pyrolysis of e-waste. However, there are less contents in this section. I suggest give more illustration, Maybe some reports, such as [Yu S, Su W, Wu D, et al. Thermal treatment of flame retardant plastics: A case study on a waste TV plastic shell sample[J]. Science of The Total Environment, 2019, 675: 651-657.Xiong J, Yu S, Wu D, et al. Pyrolysis treatment of nonmetal fraction of waste printed circuit boards: Focusing on the fate of bromine[J]. Waste Management & Research, 2020, 38(11): 1251-1258.] can help you to improve it.

Reviewer 3 Report

Please revise it according to the following comments:

      1. How did the you do quality control (QC) and quality assurance (QA) on the obtained secondary data to validate the conclusions about the topic

      2.  A comprehensive managerial insight should be provided in this paper.
